# Competencies in Basic Life Support after a Course with or without Rescue Ventilation: Historical Cohort Study

**DOI:** 10.3390/healthcare10122564

**Published:** 2022-12-17

**Authors:** Jordi Castillo, Adrián González-Marrón, Anna Llongueras, Laia Camós, Mireia Montané, Encarnación Rodríguez-Higueras

**Affiliations:** 1Universitat Internacional de Catalunya (UIC), Sant Cugat del Vallès, 08195 Barcelona, Spain; 2Sistema Emergències Mèdiques (SEM), 08908 Barcelona, Spain

**Keywords:** cardiopulmonary resuscitation, learning, simulation training, students, basic life support, hands-only

## Abstract

Background: Simplifying the international guidelines to improve skills after training and their retention over time has been one of the top priorities in recent years. The objective of our study was to compare the results of the practical skills learned during training in basic life support with and without pulmonary ventilation. Methods: This was a comparative study of historical cohorts consisting of undergraduate students in health sciences. In one cohort, rescue breathing was performed, and in the other, it was not. The same data collection instruments were used for both cohorts: a test type examination of knowledge, data from a smart mannequin and an instructor observation grid. The means of knowledge and practical skills scores collected by the mannequin were compared using independent sample t-tests. Results: 497 students were recruited without significant differences between the two cohorts. The mean scores for knowledge and skills determined by the instructor and the mannequin were statistically higher in the cohort that did not perform rescue breathing. Conclusion: Students who participated in basic life support training that did not include rescue breathing scored better than those who participated in training that included this skill. Training with only compressions simplifies the guidelines and increases learning and content retention.

## 1. Introduction

Since the beginning of modern cardiopulmonary resuscitation (CPR), around the 1960s [1,2], artificial ventilation has been an essential part of the teaching and care of sudden cardiac arrest. Lung ventilation through “mouth to mouth” or with various more invasive methods were always the most difficult and less intuitive resuscitation techniques to learn and practice and were the components that were most frequently failed [3]. In the 1990s, various investigations using simulation mannequins and using animals and humans highlighted the importance of quality chest compressions and their noninterruption during CPR. All of this generated a tendency (especially in the United States) to prioritize external cardiac compressions over ventilation. This resulted in clinical results similar to those of traditional CPR [4,5] and a simplification of the care and teaching methods. In this sense, the COVID-19 pandemic has had a decisive influence on this trend [6].

In 1993, Berg et al. [3], after their experimental study in pigs, suggested that mouth-to-mouth ventilation during CPR could be delayed up to 12 min without compromising survival or resulting in neurological alterations. Charlier et al. [7] and Baldi et al. [8] proposed simplifying both the technique and the teaching of CPR by eliminating ventilation from the international guidelines. 

After these studies, the international guidelines of the European Resuscitation Council (ERC) of 2010 [9] and 2015 [10] began to recommend teaching only chest compression use in basic life support (BLS) courses and the use of the automated external defibrillator (AED) when performed by people with little or no training in this matter.

In Europe, however, accredited training in BLS-AED continues to include rescue breathing [10], unlike in other countries [11,12]. There are few articles comparing the learning outcomes of both types of training (with and without rescue breathing).

The objective of our study was to compare the theoretical knowledge and practical skills learned after regulated training in BLS-AED with and without rescue ventilation with the aim of making the learning and improvement of the practice of CPR easier.

## 2. Materials and Methods

### 2.1. Design

This study was a historical cohort study consisting of students studying nursing, medicine and psychology at a Spanish school of medicine and health sciences who were enrolled in the first year of their degree and who completed an accredited course in BLS-AED of 4 h according to the 2015 recommendations of the ERC [13].

### 2.2. Ethical Considerations

This study was authorized by the ethics committee of the university (nº. INF-2020-07).

### 2.3. Participants and Sample Size

The first cohort (C1) (n = 255) consisted of students from the 2019-20 academic year who completed a conventional BLS-AED course. The second cohort (C2) (n = 242) consisted of students from the 2020-21 academic year who completed a BLS-AED course excluding the teaching of rescue breathing. In both cohorts, those students who had an accredited diploma approved in the previous 3 years and who were repeating the course were excluded. The only difference between the cohorts was that, in C2, the training was carried out without rescue ventilation due to the COVID-19 pandemic. All the health recommendations for the practical training were taken into account (large and airy classrooms and disinfection of the mannequins and material used) [6].

### 2.4. Intervention and Outcomes

Both trainings were given by the same 8 instructors who were accredited by the ERC and who used the same teaching methodology [10].

### 2.5. Variables

#### Dependent Variables

The students were scored at the end of the training using the following instruments: (a) A questionnaire of 10 multiple choice questions agreed to by the scientific society of the autonomous community of the study that officially certified the training. (b) An 8-item grid that evaluated the sequence of actions observed by the instructor (with a Likert scale of 0–2 with a maximum of 16 points and a minimum of 0) and was used by Castillo et al. [14] which collected the following: consciousness assessment, breathing assessment, breathing assessment time, calling of 112, requesting for AED, placement of patches, safe discharge and uninterrupted compressions. (c) The data were collected automatically by the smart mannequin Resusci Anne QCPR^®^ with SkillReporter [15] software. This was configured under the guidelines of the ERC in 2015. Additionally, in the first year (Cohort C1), the CPR 30:2 mode (hand placement, depth, recoil and frequency in compressions, as well as volume and correct ventilations) was used, and, in the second year, the “compression only” mode was used.

The independent variable was the type of training (with or without rescue ventilation), and covariates were university degree, age, sex, weight and height.

The data were collected individually at the end of training with a simulated case (which was the same for all students) of sudden cardiac arrest after the performance of 2 min of CPR.

### 2.6. Data Analysis

Qualitative variables were described as the absolute frequency (n) and percentage (%), and quantitative variables were described as the mean and standard deviation (SD). To verify the normality of the quantitative variables, the Kolmogorov–Smirnov test was used. For the comparison between groups, Student’s *t*-test of independent samples was used to compare means, and the chi-square test was used to compare proportions. The scores of the questionnaire and grid were transformed to a scale of 0 to 10 to facilitate their interpretation. The level of significance was set at 0.05. For the statistical analysis, the statistical program SPSS for Windows version 18 was used.

## 3. Results

A total of 497 students participated in the study and were divided into two cohorts of 255 and 242 students. No statistically significant differences were observed in any covariable between the cohorts (Table 1).

In Table 2, it can be observed that the means of the global scores of C2 were statistically higher than the means of the global scores of C1 both in the knowledge scores and in the data provided by the smart mannequin.

However, when comparing the skills monitored by the mannequin, statistically higher means could be observed in the chest recoil and the mean depth of compressions in C1 compared to C2 and in the frequency in C2 compared to C1.

The percentage of correct ventilations in C1 was 62.2% although only 32.1% of the ventilations provided correct volumes of 400–700 mL.

Table 3 shows that the means of the scores of all the items evaluated by the instructors were statistically higher in C1 except for those of the correct placement of the defibrillator patches and immediate compressions in which no statistically significant differences were observed between the cohorts.

## 4. Discussion

The two scientifically proven effective procedures in CPR are quality chest compressions and early defibrillation [16]. However, there are no studies that demonstrate this efficacy for rescue breathing. In fact, in teleoperator-guided CPR, ventilation is not taken into account [16].

“Mouth-to-mouth” ventilation can be an obstacle for first responders since it is an intimidating maneuver, it is difficult to remember and perform [3] and its disuse has been aggravated by COVID-19 [6]. Therefore, most potential resuscitators do not want or cannot perform conventional CPR [17], and trainers have difficulties in recommending it.

Our study suggested that training without learning ventilation yields better overall results than if it is included.

Surprisingly, the theoretical knowledge was better in the group that did not perform rescue breathing. The simplified content could be a reasonable explanation. The mannequin-reported overall data was better in the group that did not perform ventilation [18,19]. As in many other studies, ventilation was the poorest performed skill [20,21]. In our group, only 62.2% of the students performed the ventilations correctly, and only 32.1% performed them with an adequate volume.

In the data collected by the instructors, the group that learned CPR without ventilation scored statistically better in six of the eight items. Our interpretation was that the simplification of the algorithm for BLS-AED training improved the acquisition of skills. The weakest link in the survival chain obtained higher values when ventilation was not performed, and, consequently, survival could be higher.

We could not assure the differences in the quality of the students who performed the procedure. We could not retrieve their basic academic skills, and there could be potential confounding variables. This study was an observational and retrospective study, and we knew that this was a limitation in our study.

Some studies with first responders have noted better clinical outcomes without rescue ventilation [11,22]. However, the Japanese observational study by Kitamura et al. [23], which compared patient survival and neurological status at discharge, found better results in conventional outpatient CPR. The first systematic reviews and meta-analyses have already been published [5,22] that conclude that, despite having few randomized studies and despite the majority of studies being observational, there are no significant differences in the clinical outcomes between standard CPR and CPR without ventilation.

However, there are circumstances in which conventional CPR with rescue breathing is the preferred choice, such as in pediatric sudden cardiac arrest, in drowning and other forms of respiratory failure as well as in resuscitation performed by professional teams with high-quality training [16].

Perhaps we should join efforts to train the entire population in performing quality compressions without rescue breathing and leave ventilation exclusively to health personnel who have instrumentalized materials that allow them to be able to perform this maneuver effectively. In addition, the pandemic situation will make this maneuver unrecommended by the guidelines for COVID-19 [6].

However, studies that show the results of learning with or without rescue breathing are scarce due to the lack of a unified international criterion.

While they may evaluate retention in the months following the training, it is understood that outcomes are surely better when the training is easier and simpler [16].

Given that the objective of our study was in accordance with the international recommendations, future research should be aimed at carrying out this study on cardiac arrest in humans.

## 5. Conclusions

In our experience, the theoretical knowledge and practical skills acquired after a university training course in BLS-AED were better when the teaching of rescue breathing was not included. Training with only compressions may increase learning and content retention.

## Figures and Tables

**Table 1 healthcare-10-02564-t001:** Comparison of the cohorts with respect to the covariates at the beginning of the follow-up.

		Cohort 1 (n = 255)	Cohort 2 (n = 242)	*p* Value
Gender	Male	21.2% (54)	24.4% (59)	0.23 ^a^
	Woman	78.8% (201)	75.6% (183)	
Degree	Medicine	38.4% (98)	40.9% (99)	0.12 ^a^
	Nursing	34.9% (89)	39.3% (95)	
	Psychology	26.7% (68)	19.8% (48)	
Age (years)		19.6 (3.2)	19.5 (3.9)	0.76 ^b^
Weight (kg)		60.26 (9.9)	59.91 (11.5)	0.72 ^b^
Size (Cm)		167.36 (8.2)	165.22 (22.9)	0.17 ^b^

^a^ Chi-square. ^b^ Student’s *t*-test independent samples.

**Table 2 healthcare-10-02564-t002:** Comparison of the results of knowledge scores, practical skills monitored by the mannequin and the questionnaire scores of the two cohorts.

	Cohort 1 (n = 255)	Cohort 2 (n = 242)	*p* Value ^a^
CHEST COMPRESSIONS			
Correct hand placement (%)	96.5 (15.1)	97.7 (10.5)	0.29
Complete recoil (%)	80.4 (28.3)	71.1 (33.1)	0.001
Average depth in mm	50.2 (7.9)	46.5 (7.5)	0.001
Depth of 50–60 mm (%)	56.9 (35.9)	42.2 (35.2)	0.001
Compression frequency (comp/min)	112.7 (16.5)	115.6 (9.9)	0.018
Frequency of 100–120 (%)	52.3 (35.4)	61.1 (32.9)	0.007
VENTILATION			
Correct ventilation (%)	62.2 (31.5)		
Average volume (ml)	652.8 (313.6)		
Volume > 700 mL (%)	42.7 (36.6)		
Volume of 400–700 mL (%)	32.1 (29.3)		
Volume < 400 mL (%)	17.7 (24.9)		
GLOBAL MANNEQUIN SCORE (%)	60.5 (24.1)	66.1 (27.1)	0.014
KNOWLEDGE SCORE	7.96 (1.4)	8.34 (1.5)	0.005

The results are shown as the mean and standard deviation. ^a^ Student’s *t*-test of independent samples.

**Table 3 healthcare-10-02564-t003:** Scores from the practical BLS-AED algorithm assessment of the instructors’ observations.

	Cohort 1 (n = 255)	Cohort 2 (n = 242)	*p* Value ^a^
Consciousness assessment Breathing assessment Breathing assessment time	9.1 (2)	9.8 (1)	0.001
	7.9 (3.1)	8.5 (2.4)	0.001
	7 (3.2)	8.4 (2.6)	0.001
Request for AED Calling of 112	8.5 (3.4)	9.5 (1.6)	0.001
	8.3 (3.2)	9.5 (1.8)	0.001
Correctly applied AED patches Safe discharge Immediate compressions GLOBAL INSTRUCTIONAL ASSESSMENT	9.5 (2.2)	9.6 (1.9)	0.59
	6.6 (3.9)	7.6 (3.45)	0.004
	8.1 (2.8)	8.3 (2.4)	0.26
	8.1 (2.4)	8.9 (2.1)	0.001

Scores over 10 points were included within the mean (standard deviation). AED: Automated external defibrillator. ^a^ Student’s *t*-test of independent samples.

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
