# Peer review of "Competencies in Basic Life Support after a Course with or without Rescue Ventilation: Historical Cohort Study"

_healthcare, 2022, doi:10.3390/healthcare10122564_

Round 1

Reviewer 1 Report

The objective of the authors  was to compare the results on practical skills during training in basic life support with or without pulmonary ventilation. Simplifying the international guidelines to improve skills after training and their retention over time has been one of the top priorities in recent years.

The authors proposed a comparative study of historical cohorts with undergraduate students in health sciences. In one cohort, rescue breathing was performed, and in the other, it was not. The same data collection instruments were used for both cohorts: test-type examination of knowledge, data from a smart mannequin and an instructor observation grid. The means of the knowledge and practical skills scores collected by the mannequin were compared using independent samples t-tests. A total of 497 students participated in the study, without significant differences between the two cohorts. The mean scores for knowledge and skills determined by the instructor and the mannequin were statistically higher in the cohort that did not perform rescue breathing.

The authors concluded that: (a) Students who participate in basic life support training that does not include rescue breathing scored better than those who participate in training excluding this skill. (b) Training with only compressions simplifies the guidelines and increases learning and content retention.

The ms needs substantial improvements.

These are my comments:

1)      Rewrite the abstract. It must better summarize the sections. Now it directly starts with the aim.

2)      Expand the purpose. Now it is a bit cryptic “The objective of our study is to compare the theoretical knowledge and practical skills after regulated training in BLS-AED with or without rescue ventilation.”

3)      M&M. There many paragraphs based on one sentence. Please rewrite. In the present form is sounds too fragmented.

4)      Results must be expanded and rewritten with more care and details. See for example the first two sentences  “This section may be divided by subheadings. It should provide a concise and precise”. Table 3 is not standard

5)      Discussion. The sentence “we should join efforts to train the entire population in performing quality compressions without rescue brea..” is not clear.

6)      Discussion. Add the limitations.

7)      Conclusions are poor

Author Response

Point 1: Various grammar and layout issues:

  • Line 11: of practical skills

 Response 1:  we have changed “on practical skills by“ “of practical skills”

Point 2:

  • line 16: a test type

Response 2: we have changed “test-type” by “a test type”

Point 3:

  • line 17: remove article before knowledge

Response 3:  

  • we have changed “of the knowledge” by “of knowledge”

 Point 4:

  • line 29: beginning without s

Response 4: 

  • we have changed “beginnings” by “beginning”

Point 5:

  • line 32: “most difficult”…i.e. less intuitive

Response 5: Ventilation techniques have to be learned on a basis protocol and are difficult to learn but I understand that they are also less intuitive.  We have added your suggestion on the manuscript

Point 6:

  • lines 43-44: rephrase

 Response 6: We have completely rephrased lines 43 and 44. We have changed reference number 8 and we think that now it makes much sense.

 Point 7:

  • line 149: surprisingly

Response 7:

We agree with the reviewer that this sentence doesn’t make sense. We have rewrite it beginning by “susprinsingly”

Surprisingly the theoretical knowledge is better in the group that did not perform rescue breathing. The simplified content could be a reasonable explanation.

 Point 8: Rephrase conclusions

Response 8: -     we have changed “we concluded that” by “In our experience….”

Point 9: Rephrase conclusions

  • Training with only compressions may increase learning and content retention.

Response 9: According to your suggestion, we added the above sentence.

Reviewer 2 Report

Various grammar and layout issues:
-        Line 11: of practical skills
-        line 16: a test type
-        line 17: remove article before knowledge
-        line 29: beginning without s
-        line 32: “most difficult”…i.e. less intuitive
- lines 43-44: rephrase
-        line 149: surprisingly
Rephrase conclusions
In our experience….

Training with only compressions may increase learning and content retention…

Author Response

Response to Reviewer 1 Comments

Point 1: Rewrite the abstract. It must better summarize the sections. Now it directly starts with the aim.

Response 1: According to the reviewer comments, we have summarized the sections and changed a sentence in order to begin with a background.

Point 2: Expand the purpose. Now it is a bit cryptic “The objective of our study is to compare the theoretical knowledge and practical skills after regulated training in BLS-AED

Response 2: At the end of the introduction we have added “the aim of make easer the learning and improve the practice of CPR”with or without rescue ventilation.”

Point 3: M&M. There many paragraphs based on one sentence. Please rewrite. In the present form is sounds too fragmented.

Response 3: Done as suggested

Point 4: Results must be expanded and rewritten with more care and details. See for example the first two sentences “This section may be divided by subheadings. It should provide a concise and precise”. Table 3 is not standard

Response 4.  The first two sentences of the results were written by error from the template. We apologize for that. We have deleted it. We think that results are enough explained on the text and tables. More details would be redundant. We have modified the table 3 according to the template.

Point 5: Discussion. The sentence “we should join efforts to train the entire population in performing quality compressions without rescue brea..” is not clear.

Response 5: We don’t understand why this sentence is not clear. We mean that if International Guidelines would recommend to teach and practice CPR without ventilation, it will be easier to train the general population.

Point 6:   Discussion. Add the limitations.

Response 6: Another reviewer makes the same suggestion. We have added a limitation to the study.

“we cannot assure the differences in the quality of the students who performed the procedure. We cannot retrieve the basic academic skills and there could be potential confounding variables. It is an observational and retrospective study and we know that it is a limitation of our study”.

Point 7: Conclusions are poor

Response 7: Another reviewer makes the same suggestion. We have added another conclusion. “Training with only compressions may increase learning and content retention”. We think that based on our results nothing else can be concluded”

Reviewer 3 Report

This is a paper on a qualitative evaluation of BLS using mannequins and the quality of chest compressions is shown to be better in the group without ventilation. The authors states that a simplified algorithm led to improve the acquisition of the skills. Background comparisons between the two groups show no differences in gender, degree, age, height, or weight. However, there is no mention of the differences in the quality of the students who performed the procedure. Because no objective assessment of the competence of the students in both groups has been made, there may be differences in the basic academic skills and understanding of resuscitation between two groups. As shown in table 3, all the scores were inferior in cohort 1. I think that the students in cohort 2 may have been superior and had higher basic academic skills than the students in cohort 1. If the students in cohort 2 had performed ventilation, a more ideal technique might have been performed. In accordance with this hypothesis, the conclusions of this paper may be meaningless. The authors should respond on this point.

Author Response

Point 1: This is a paper on a qualitative evaluation of BLS using mannequins and the quality of chest compressions is shown to be better in the group without ventilation. The authors states that a simplified algorithm led to improve the acquisition of the skills. Background comparisons between the two groups show no differences in gender, degree, age, height, or weight. However, there is no mention of the differences in the quality of the students who performed the procedure. Because no objective assessment of the competence of the students in both groups has been made, there may be differences in the basic academic skills and understanding of resuscitation between two groups. As shown in table 3, all the scores were inferior in cohort 1. I think that the students in cohort 2 may have been superior and had higher basic academic skills than the students in cohort 1. If the students in cohort 2 had performed ventilation, a more ideal technique might have been performed. In accordance with this hypothesis, the conclusions of this paper may be meaningless. The authors should respond on this point.

Response 1: We absolutely agree with the reviewer that we cannot assure the differences in the quality of the students who performed the procedure. We cannot retrieve the basic academic skills and there could be potential confounding variables. It is an observational and retrospective study and we know that it is a limitation of our study.

Nevertheless we have found in cohort 1 group (the one of ventilation) better skills during the performance of cardiac compressions. So we could thought that this group has a better understanding of resuscitation. This is just a suspicion and we cannot pull out conclusions from it.

We will add all this as a limitation in the discussion.

Thank you very much for your observations

Round 2

Reviewer 1 Report

N/A

Reviewer 3 Report

The items I pointed out have been appropriately corrected and comments have been added as Limitation.